

# BMSC-derived exosomal miR-27a-3p and miR-196b-5p regulate bone remodeling in ovariectomized rats

Guohua Lai[1,2], Renli Zhao[1,2], Weida Zhuang[1,2], Zuoxu Hou[1], Zefeng Yang[1,2], Peipei He[1,2], Jiachang Wu[1,2] and Hongxun Sang[1,2]

[1] Department of Orthopedics, Shenzhen Hospital, Southern Medical University, Shenzhen, China
[2] The Third School of Clinical Medicine, Southern Medical University, Guangzhou, China

## ABSTRACT

**Background**. In the bone marrow microenvironment of postmenopausal osteoporosis (PMOP), bone marrow mesenchymal stem cell (BMSC)-derived exosomal miRNAs play an important role in bone formation and bone resorption, although the pathogenesis has yet to be clarified.

**Methods**. BMSC-derived exosomes from ovariectomized rats (OVX-Exo) and sham-operated rats (Sham-Exo) were co-cultured with bone marrow-derived macrophages to study their effects on osteoclast differentiation. Next-generation sequencing was utilized to identify the differentially expressed miRNAs (DE-miRNAs) between OVX-Exo and Sham-Exo, while target genes were analyzed using bioinformatics. The regulatory effects of miR-27a-3p and miR-196b-5p on osteogenic differentiation of BMSCs and osteoclast differentiation were verified by gain-of-function and loss-of-function analyses.

**Results**. Osteoclast differentiation was significantly enhanced in the OVX-Exo treatment group compared to the Sham-Exo group. Twenty DE-miRNAs were identified between OVX-Exo and Sham-Exo, among which miR-27a-3p and miR-196b-5p promoted the expressions of osteogenic differentiation markers in BMSCs. In contrast, knockdown of miR-27a-3p and miR-196b-5p increased the expressions of osteoclastic markers in osteoclast. These 20 DE-miRNAs were found to target 11435 mRNAs. Gene Ontology and Kyoto Encyclopedia of Genes and Genomes pathway analyses revealed that these target genes were involved in several biological processes and osteoporosis-related signaling pathways.

**Conclusion**. BMSC-derived exosomal miR-27a-3p and miR-196b-5p may play a positive regulatory role in bone remodeling.

Corresponding authors
Jiachang Wu, wujiachang1982@163.com
Hongxun Sang, hxsang@smu.edu.cn

# INTRODUCTION

Osteoporosis is one of the most common diseases caused by abnormal bone metabolism and is characterized by reduced bone mass and altered microarchitecture, which increases the susceptibility to fractures (*Chen et al., 2015*; *Yang et al., 2019*; *Marini, Cianferotti & Brandi, 2016*). With the issue of increased population aging, the incidence of osteoporosis and secondary fracture has also risen rapidly, increasing the economic burden of families

and society (*Abrahamsen, Skjødt & Vestergaard, 2019*; *Lizneva et al., 2018*). Osteoporosis is most common in postmenopausal women with estrogen deficiency due to the imbalance between bone formation by osteoblasts and bone resorption by osteoclasts (*Manolagas, 2000*). Bone marrow mesenchymal stem cells (BMSCs) are one of the sources of osteoblasts, which play a major role in new bone formation (*Marie & Kassem, 2011*), while osteoclasts are the main cells responsible for bone resorption (*Wang et al., 2021*). Studies have shown that exosomes play an important role in the communication between osteoblasts and osteoclasts and affect the progression of osteoporosis, in which microRNAs (miRNAs) exhibit a major effect, but the mechanism has yet to be clarified (*Sun et al., 2016*; *Wang et al., 2021*).

MiRNAs are ~22-nucleotide noncoding RNAs that regulate gene expression at the posttranscriptional level by targeting mRNA (*Li et al., 2016*; *Ma et al., 2019*). Abnormal expression of miRNAs is involved in a variety of bone metabolic processes, including proliferation and differentiation of osteoblasts and osteoclasts (*Sun et al., 2016*). For example, miR-214 levels negatively correlate with reduced bone formation in both human and mouse bone specimens. miR-214 inhibits osteoblast activity and matrix mineralization *in vitro* and inhibits bone formation *in vivo* (*Wang et al., 2013*). However, most previous studies on miRNA have focused on intracellular, and little attention has been paid to the role of miRNA in the extracellular environment. Recent studies have shown that miRNAs facilitate intercellular communication, such as transducing signals through exosomes (*Montecalvo et al., 2012*; *Mittelbrunn et al., 2011*; *Valadi et al., 2007*).

Exosomes are extracellular vesicles, approximately 40–120 nm in diameter, that contain a variety of proteins and RNAs and are associated with cancer, immune diseases and skeletal disorders (*Ohno et al., 2013*; *Montecalvo et al., 2012*; *Xu et al., 2018a*; *Xu et al., 2018b*; *Sheinerman et al., 2013*). Interestingly, exosomes can protect RNA from the degradation of RNase because of their unique membrane structure (*Zhang & Jin, 2020*). BMSC-derived exosomal miRNA plays an important role in osteoblast and osteoclast differentiation and the development of osteoporosis. For example, up-regulation of BMSC-derived exosomal miR-150-3p promotes osteoblast proliferation and differentiation *in vitro* and attenuates the osteoporotic phenotype in ovariectomized rats *in vivo* (*Qiu et al., 2021*). MSCs from ankylosing spondylitis patients (ASMSCs) exhibit a stronger ability to inhibit osteoclastogenesis than MSCs from healthy donors and miR-4284 is decreased in ASMSCs. MiR-4284 can regulate MSC-mediated inhibition of osteoclastogenesis (*Liu et al., 2019*). Therefore, the miRNAs transferred by exosomes to recipient cells may exert a significant effect on cellular functions. To understand the role of exosomes in postmenopausal osteoporosis (PMOP), an extensive study on exosomal miRNAs is imperative.

Ovariectomy is the most common method used in the PMOP model. Typically, ovariectomized (OVX) rats were used to establish the PMOP rat model, while sham-operated (Sham) rats were used as non-osteoporotic controls. Researchers often use micro-computed tomography (micro-CT) to scan the distal femur or lumbar spine of rat models and analyze the bone microarchitecture of region of interest (ROI) by bone mineral density (BMD), bone trabecular number (Tb.N), bone trabecular spacing (Tb.Sp), bone

trabecular morphology factor (Tb.Pf), bone volume percentage (BV/TV), and bone surface density (BS/TV) to assess the osteoporosis of the model.

Osteogenic differentiation and osteoclast differentiation are the two main phenotypes in bone remodeling studies. Osteogenic differentiation ability is commonly measured by alkaline phosphatase (ALP) activity, alizarin red staining, and mRNA and protein expression of osteogenic markers, such as osteocalcin (OCN), osterix (OSX), osteopontin (OPN), and runt-related transcription factor 2 (RUNX2), *etc* (*Zhang et al., 2021*; *Zhang et al., 2020*). Furthermore, osteoclast differentiation ability is commonly assayed by the number of tartrate resistant acidic phosphatase (TRAP)-positive cells and the expression of osteoclastic markers, such as TRAP, cathepsin K (CTSK), and nuclear factor of activated T-cell cytoplasmic 1 (NFATc1) (*Chen et al., 2014*; *Yu et al., 2021*).

This study focused on identifying the differentially expressed miRNAs (DE-miRNAs) of BMSC-derived exosomes from OVX rats (OVX-Exo) and Sham rats (Sham-Exo) using next-generation sequencing (NGS) and validating the regulatory roles of candidate miRNAs on BMSCs osteogenic differentiation and osteoclast differentiation. Thus, our study provides a potential therapeutic approach for PMOP.

## MATERIAL AND METHODS

### Experimental animals

A total of 18 Sprague-Dawley rats (female, 12 weeks old, weighing 220 ± 10 g) were purchased from Beijing Vital River Laboratory Animal Technology Co., Ltd. (Beijing, China). All experimental animal procedures were approved by the Animal Ethics Committee of Shenzhen Hospital of Southern Medical University, China (approval No. 2021-0057). The rats were maintained on a 12-h light/dark cycle at 22 °C with 50–55% humidity, and got free access to food and water. After one week of acclimatization, the rats were equally and randomly assigned to the OVX and Sham groups (9 rats per group). The rats were anesthetized by an intraperitoneal injection of 2% 2,2,2-tribromoethanol (Macklin, Shanghai, China). Then, bilateral ovaries were removed from the OVX rats, while peripheral ovarian fat was removed from the Sham rats, as described previously (*Wu et al., 2021*). After completion of operation the resuscitated rats were kept in the same conditions as before operation. To evaluate the rat PMOP model, rats were euthanized by cervical decoupling 12 weeks after the operation, and the fourth lumbar vertebrae (L4) were scanned using micro-CT (SkyScan 1176; Bruker, Kontich, Belgium) and analyzed for bone microstructure. The parameters were as follows: voxel size 10.139475 μm, medium resolution, 85 kV, 200 μA, one mm Al filter and integration time 384 ms. The data were analyzed using the manufacturer's evaluation software, including SkyScan NRecon (version 1.7.4.2), CTvox (version 3.3.0) and CTAn (version 1.18.8.0).

### Culturing and identification of BMSCs

BMSCs were isolated from the femur and tibia of OVX and Sham rats and cultured in growth medium, as described previously (*Xu et al., 2018a*; *Xu et al., 2018b*). Briefly, the bilateral femurs and tibia were harvested under sterile conditions, and the bone marrow was collected. BMSCs were cultured in 10-mm Petri dish and replaced with the growth

medium every other day. The growth medium included $\alpha$-minimum essential medium ($\alpha$-MEM), 10% fetal bovine serum (FBS), and 1% penicillin and streptomycin solution (Gibco, Carlsbad, CA, USA). BMSCs were passaged when the cell density reached 80–90% confluency.

Flow cytometric characterization was performed as described previously (*Wu et al., 2021*). Briefly, the passage 3 BMSCs were digested with 0.25% trypsin and suspended to a concentration of $1 \times 10^6$ cells/mL with cell staining buffer (BioLegend, San Diego, CA, USA). Next, the cells were incubated with antibodies against CD29-phycoerythrin (PE; # 102207), CD90-PE (#202207), CD11b-PE (#201807), or CD90-PE (#202523, BioLegend, San Diego, CA, USA) in the dark at 4 °C for 45 min. A tube of cells suspended in PBS was set as the control group. Subsequently, the cells were collected by centrifugation (4 °C, 150 $\times$ *g*, 5 min), and the cells were analyzed by flow cytometry on a Sony SA3800 apparatus (Sony Biotechnology, Tokyo, Japan).

For osteogenic differentiation of BMSCs, the cells were cultured in osteogenic induction medium (Cyagen Biosciences, Guangzhou, China) for 14 days and stained with alizarin red. For adipogenic differentiation, the cells were stained with Oil Red O after 3 weeks of culturing in adipogenic induction medium (Cyagen Biosciences).

## Osteoclast culture and differentiation induction

For the isolation of bone marrow-derived macrophages (BMMs), bone marrow cells (BMCs) were lysed with erythrocyte lysate buffer (Leagene, Beijing, China) and centrifuged at 1200 rpm/min for 5 min. After removal of the supernatant, BMCs were cultured overnight in complete medium ($\alpha$-MEM + 10% FBS), and then the unadhered cells were collected and cultured for 2 days in complete medium supplemented with 30 ng/mL macrophage colony stimulating factor (M-CSF) (PeproTech, Hamburg, Germany). BMMs were then inoculated into 96-well culture plates (8000 cells/well) and cultured in complete medium containing 30 ng/mL M-CSF and 50 ng/mL receptor activator ligand of NF-kB (RANKL) (PeproTech) for 4 days (the medium was changed every 2 days). Tartrate-resistant acid phosphatase (TRAP) activity was detected with a TRAP Stain Kit (Sigma-Aldrich, St. Louis, MI, USA) according to the manufacturer's protocol, and the mRNA of osteoclastic markers were detected by real-time quantitative reverse transcription-PCR (qRT-PCR).

## Isolation and characterization of BMSC-derived exosomes

When passage 3 BMSCs reached a confluency of 80–90%, the culture medium without exosomes was replaced, and the supernatant was collected after 2 days. The exosomes in the supernatant of BMSCs were extracted by differential ultracentrifugation (Rotor: SW *32 Ti*, Beckman Coulter, Brea, CA, USA). Briefly, the supernatant was collected by centrifugation at 300, 2000, and 10,000 $\times$ *g* for 10, 10, and 30 min to remove the cells and cell debris. Next, protein contaminants were removed by ultracentrifugation of the supernatant at 110,000 $\times$ *g* for 70 min twice, and the precipitate was resuspended in PBS buffer. Subsequently, exosomes in the precipitate were resuspended in 100 µL PBS and stored at −80 ° C (*Théry et al., 2006*; *Chen et al., 2019*).

The approximate concentration and diameter of the exosomes were measured by nanoparticle tracking analysis (NTA) using a ZetaView Particle Metrix (Particle Metrix,

Meerbusch, Germany). The shape and size of the exosomes were observed by transmission electron microscopy (TEM) (FEI Tecnai G2 Spirit BioTwin; FEI, Hillsboro, OR, USA). The exosome positive markers (CD81, Hsp70, and TSG101) and purity control (Calnexin) were detected by Western blotting.

## Isolation of exosomal total RNA and miRNA sequencing

BMSC-derived exosomal total RNA was extracted using TRIzol (Thermo Fisher Scientific, Waltham, MA, USA) according to the manufacturer's instructions. The integrity of total RNA was analyzed by an Agilent 2100 bioanalyzer (Agilent Technologies, Santa Clara, CA, USA), and RNA integrity number (RIN) >7 was used for NGS. The concentration of total RNA was detected by NanoDrop$^{TM}$ 2000 (Thermo Fisher Scientific) (*Pang et al., 2020*). As described previously, miRNA sequencing was performed on the Illumina HiSeq 2500 platform from LC Sciences (Hangzhou, China) (*Pang et al., 2019*). The raw reads of each sample were internally programmed by ACGT101-miR, and clean reads were obtained by removing the adapter dimers, junk, low complexity, common RNA families (rRNA, tRNA, snRNA, and snoRNA), and repetition. Then, unique sequences of 18–26 nucleotides were mapped to the precursors of *Rattus norvegicus* in miRBase 22.0 to identify the known and novel miRNAs. L/R ± $n$ indicated that the detected miRNA sequence was n bases more/less than the known miRNAs on the left/right side. The read counts corresponding to tags per million counts (TPM) ratio was used to normalize the expression levels of miRNAs. DE-miRNAs were identified by fold-change ≥2 or fold-change ≤0.5 and $P \leq 0.05$.

## Target gene prediction, GO and pathway enrichment analysis

To study the possible functions of DE-miRNAs, miRanda and TargetScan algorithms were used to predict their target genes that were subsequently removed with a context score percentile <50 in the TargetScan algorithm and a max energy value > −10 in the miRanda algorithm. Finally, those found at the intersection of the two algorithms were considered the final target genes of DE-miRNAs. The Gene Ontology (GO) project (http://www.geneontology.org) was used to analyze the gene function of the target genes of DE-miRNAs. We identified the significant pathways associated with enriched DE-miRNAs according to KEGG analysis (http://www.genome.jp/kegg). Fisher's exact test and $\chi^2$ test were used to classify the GO categories and KEGG pathways, and the false discovery rate (FDR) was calculated to correct the $P$-value. The thresholds of $P < 0.05$ and FDR < 0.05 were used to select the significant GO categories and KEGG pathways.

## Exosome uptake assay

OVX-Exo and Sham-Exo were incubated with PKH26 (Sigma) for 5 min, and then the fluorescently labeled exosomes were centrifuged at 10,000 × $g$ for 30 min and the supernatant was collected to remove the unbound dye. Subsequently, the exosomes were collected by 110,000 × $g$ ultracentrifugation for 70 min. BMMs were cocultured with PKH26-labeled exosomes for 24 h and fixed with 4% paraformaldehyde. In the blank control group, exosomes were replaced with an equal volume of PBS. Cell cytoskeleton and nucleus were labeled with phalloidine and DAPI (Solarbio, Beijing, China), respectively, and observed under an Olympus FV1200 confocal microscope (Olympus, Tokyo, Japan).

## Cell transfection

When the confluence of BMSCs and BMMs reached 50–60%, cells were transfected with mimic-NC, miR-27a-3p mimic, miR-196b-5p mimic, inhibitor-NC, miR-27a-3p inhibitor or miR-196b-5p inhibitor (RiboBio, Guangzhou, China) using Lipofectamine$^{TM}$ RNAiMAX Reagent (Invitrogen, Carlsbad, CA, USA) according to the manufacturer's instructions. The final concentration of miRNA mimic and mimic-NC were 100 nM, and miRNA inhibitor and inhibitor-NC were 200 nM. After 12 h of transfection, the transfection reagent was replaced with osteogenic induction medium or osteoclast induction medium. After 48 h of induction, mRNAs and proteins of osteogenic or osteoclastic markers were detected by qRT-PCR and Western blot, respectively.

## qRT-PCR

Total RNA of cells and exosomes was extracted by TRIzol (Thermo Fisher Scientific). When the exosomes were fully lysed, 1 pmol of cel-miR-39-3p standard RNA was added. For miRNA detection, total RNA was reverse transcribed and amplified using the miDETECT A Track$^{TM}$ miRNA qRT-PCR Starter Kit (RiboBio). For mRNA detection, total RNA was reverse transcribed with a TransScript All-in-One First Strand cDNA Synthesis SuperMix kit (TransGen Biotech, Beijing, China). PerfectStart Green qPCR SuperMix kit (TransGen) was used for quantitative real-time PCR (qPCR) on a LightCycler 96 instrument (Roche, Basel, Switzerland). The relative quantification was carried out by the $2^{-\Delta\Delta CT}$ method. Cel-miR-39-3p was used as an external control for the detection of exosomal miRNAs, and U6 and $\beta$-actin were used as internal controls for cellular miRNA and mRNA detection, respectively. Cel-miR-39-3p standard RNA (Cat. miRB0000010), cel-miR-39-3p primer (Cat. miRA0000010) and U6 primer (Cat. miRAN0002) were purchased from RiboBio. Primer sequences for qRT-PCR are listed in Table S1.

## Western blot

Total proteins were extracted from the cells and exosomes using the Whole Cell Lysis Assay (KeyGen Biotech, Nanjing, China), and the protein concentration was determined by the FD$^{TM}$ BCA Protein Quantitative Kit (Fdbio Science, Hangzhou, China). The lysates were separated by 10% SDS-PAGE (Fdbio Science), and the proteins were transferred to a polyvinylidene difluoride membrane (Millipore, Billerica, MA, USA). Then, the blotting membrane was blocked with 5% milk in tris buffered saline with Tween 20 (TBST; Fdbio Science) at room temperature for 1 h, followed by probing with rabbit anti-CD81 (ab109201; Abcam, Cambridge, UK; 1:1,000), anti-TSG101 (ab125011; Abcam; 1:1,000), anti-Hsp70 (ab181606; Abcam; 1:1,000), anti-Calnexin (ab22595, Abcam; 1:1,000), anti-ALP (DF6225; Affinity Biosciences, Jiangsu, China; 1:500), anti-OCN (sc-390877; Santa Cruz Biotechnology, Santa Cruz, CA, USA; 1:100), anti-OSX (ab209484; Abcam; 1:1,000), anti-RUNX2 (ab236639; Abcam; 1:1,000) or anti-beta-actin (KC-5A08; Kangcheng Biotech, Shanghai, China; 1:10,000) antibodies, at 4 ° C overnight. Subsequently, the membranes were incubated with horseradish peroxidase (HRP)-coupled goat anti-rabbit IgG H&L (ab205718; Abcam; 1:10,000) or goat anti-mouse IgG H&L (HRP) (ab205719; Abcam; 1:10,000). Signals were detected with the Immobilon Western Chemiluminescent HRP substrate (Millipore) and recorded on an X-ray film (Estman Kodak Company, XBT-1).

## Statistical analysis

Statistical analysis was carried out using SPSS 25.0 software (IBM Corp., Armonk, NY, USA.), and $P < 0.05$ indicated statistical significance. Measurement data were expressed as the mean $\pm$ standard deviation (SD). Normally distributed data and homogeneity of variance between the two groups were compared using unpaired t-tests.

# RESULTS

## Evaluation of the osteoporotic rat model and identification of BMSCs

Analysis of morphological data showed that BMD and Tb. N of the fourth lumbar vertebrae were significantly decreased in OVX rats compared to Sham rats, BV/TV and BS/TV showed a decreasing trend, while Tb. Sp and Tb. Pf were significantly increased (Fig. 1A). The micro-CT images showed that the trabecular bone of fourth lumbar vertebrae decreased significantly in OVX rats compared to Sham rats (Fig. 1B). After 2 weeks of osteogenic differentiation of BMSCs, a large number of red nodules could be seen by alizarin red staining (Fig. 1C). 3-week after induction of adipogenic differentiation, Oil Red O staining showed a large number of red lipid droplets (Fig. 1D). Flow cytometry analysis showed that the expression trend of the surface markers of MSCs was similar in the OVX and Sham rats. Specifically, CD29 and CD90 were highly expressed, while CD45 and CD11b had relatively low expressions. The expressions of CD29 (99.9%), CD90 (99.9%), CD45 (2.50%) and CD11b (1.58%) in OVX rats and their corresponding expressions in Sham rats (100%, 100%, 1.94% and 1.09%, respectively) were consistent with MSC characteristics (Fig. 1E) (*Su et al., 2019*).

## Characterization and miRNA expression of BMSC-derived exosomes

TEM analysis showed that OVX-Exo and Sham-Exo were membrane microcapsules with a size of 40–120 nm (Fig. 2A). Western blot analysis indicated that the exosome markers CD81, HSP70 and TSG101 were highly expressed in OVX-Exo and Sham-Exo but were at low levels or absent in the cell lysates, in contrast to Calnexin (Fig. 2B). In addition, NTA confirmed that OVX-Exo were enriched in the range of 40–200 nm with a peak value of 107 nm (Fig. 2C), which were similar with those in Sham-Exo (Fig. 2D), consistent with previous reports (*Kalluri & LeBleu, 2020*).

## OVX-Exo promoted the expression of osteoclastic markers compared to Sham-Exo

After treatment with M-CSF and RANKL for 4 days, the number of TRAP-positive cells (Figs. 3A and 3B) and the mRNA expression of osteoclastic markers (Fig. 3C), such as tartrate resistant acidic phosphatase (ACP5), CTSK, and NFATc1, were significantly elevated compared with the control group. To explore the effects of Sham-Exo and OVX-Exo on osteoclast differentiation, the exosomes were cocultured with BMMs and the culture medium (complete medium containing M-CSF, RANKL and OVX-Exo or Sham-Exo) was changed every 2 days. After 4 days of coculturing, the mRNA expression of osteoclastic markers were remarkably elevated in the OVX-Exo-treated group compared with the Sham-Exo-treated group (Fig. 3D). In addition, after 24 h of coculturing, confocal

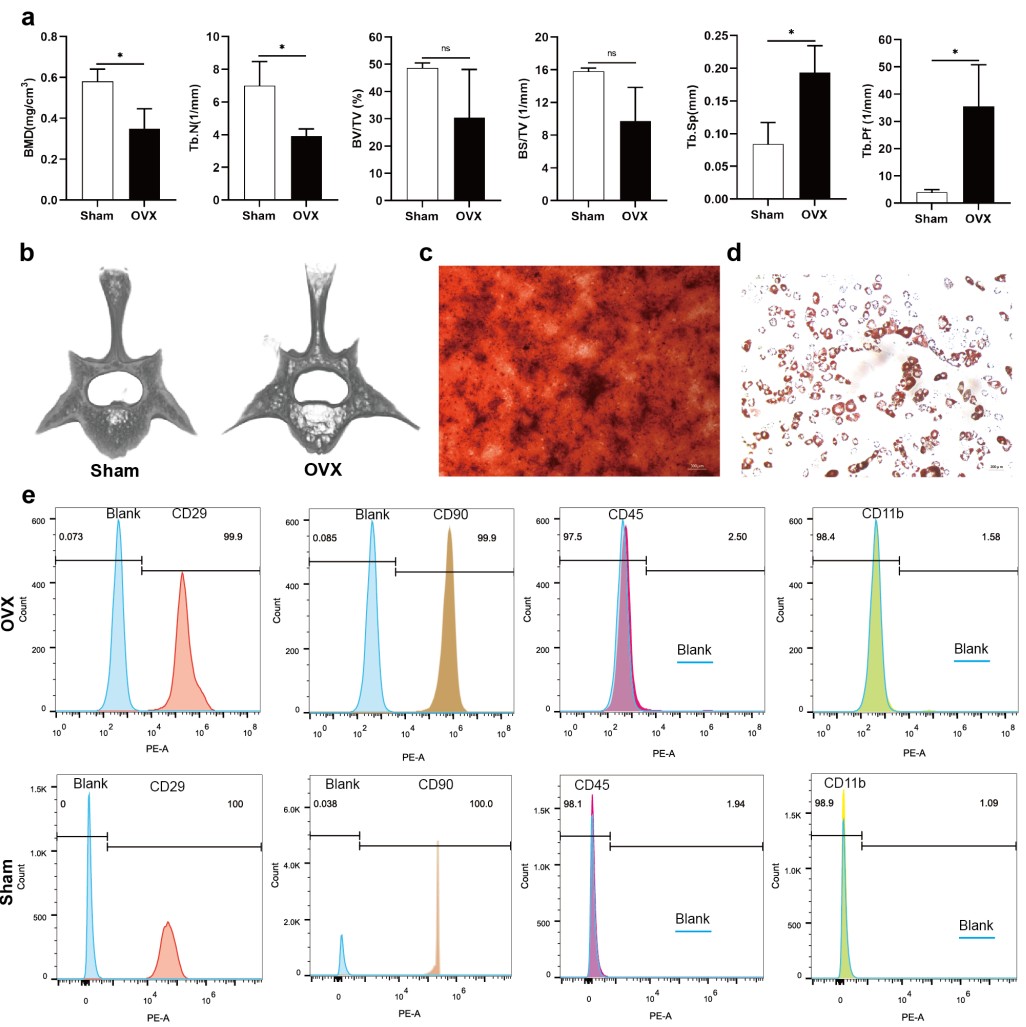

**Figure 1** **Evaluation of the osteoporotic rat model and identification of BMSCs.** (A) BMD, Tb. N, BV/TV, BS/TV, Tb. Sp and Tb. Pf of the fourth lumbar vertebrae from Sham and OVX rats. (B) Micro-CT three-dimensional view of the fourth lumbar vertebrae from Sham and OVX rats. (C) Evaluation of osteogenic differentiation of BMSCs by alizarin red staining (scale bars = 300 µm). (D) Evaluation of adipogenic differentiation of BMSCs by Oil Red O staining (scale bars = 200 µm). (E) The expressions of CD29, CD90, CD45 and CD11b in BMSCs are analyzed by flow cytometry. Data are presented as the mean ± SD, $n = 3$ rats/group. * $P < 0.05$, ns: not significant.

microscopy images showed that Sham-Exo and OVX-Exo were incorporated into the cells, while no red fluorescence signal was detected in the cytoplasm of PBS group (Fig. 3E). Which molecule inside the exosomes was at play? We explored this query in the follow-up experiment.

## MiRNA expression of OVX-Exo and Sham-Exo

NGS revealed that a total of 708 miRNAs were identified in OVX-Exo and Sham-Exo (Table S2). Among these, 20 miRNAs (14 known and 6 novel) were significantly and differentially expressed ($P < 0.05$) between OVX-Exo and Sham-Exo (Table 1). The

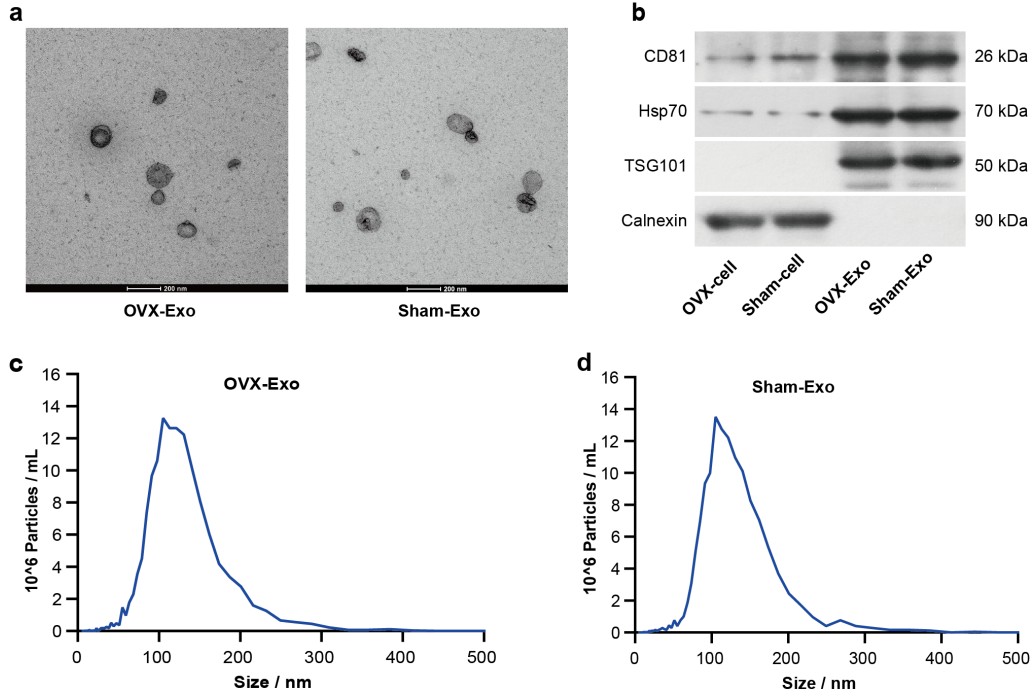

**Figure 2 Characterization of BMSC-derived exosomes.** (A) Transmission electron microscopy image showing the shape and size of OVX-Exo and Sham-Exo (scale bars = 200 nm). (B) Western blot indicats the expression of CD81, Hsp70, TSG101 and Calnexin in cellular and exosomal lysates of BMSCs. (C and D) Particle size distribution of OVX-Exo and Sham-Exo using nanoparticle tracking analysis.

hierarchical clustering illustrated the DE-miRNA profiles (Fig. 4A). The volcano map reflected the distribution of all DE-miRNAs, with rno-miR-27a-3p and rno-miR-196b-5p specially annotated (Fig. 4B).

## Target gene prediction and bioinformatics analysis of DE-miRNAs

In this study, 20 DE-miRNAs targeted 5515 mRNAs (Table S3), with rno-miR-27a-3p having the highest number of target genes (1027 genes) (Fig. 5A), which *Adcyap1r1* and *Bdnf* were target genes of 16 DE-miRNAs (Fig. 5B). These target genes were significantly ($p < 0.05$) enriched in 1200 GO terms (835 biological processes, 161 cellular components, and 204 molecular functions) (Table S4). Fig. 5C lists the 10 most enriched GO terms according to $p$-value in terms of biological processes, cellular components and molecular functions. KEGG pathway analysis revealed 139 significantly different ($P < 0.05$) pathways between the OVX-Exo and Sham-Exo (Table S5). The 20 most significantly enriched KEGG pathways of target mRNAs of DE-miRNAs were shown in Fig. 5D. These target genes were enriched in several osteoporosis-related pathways, such as HIF-1, MAPK, Hippo, Ras, and p53 signalings.

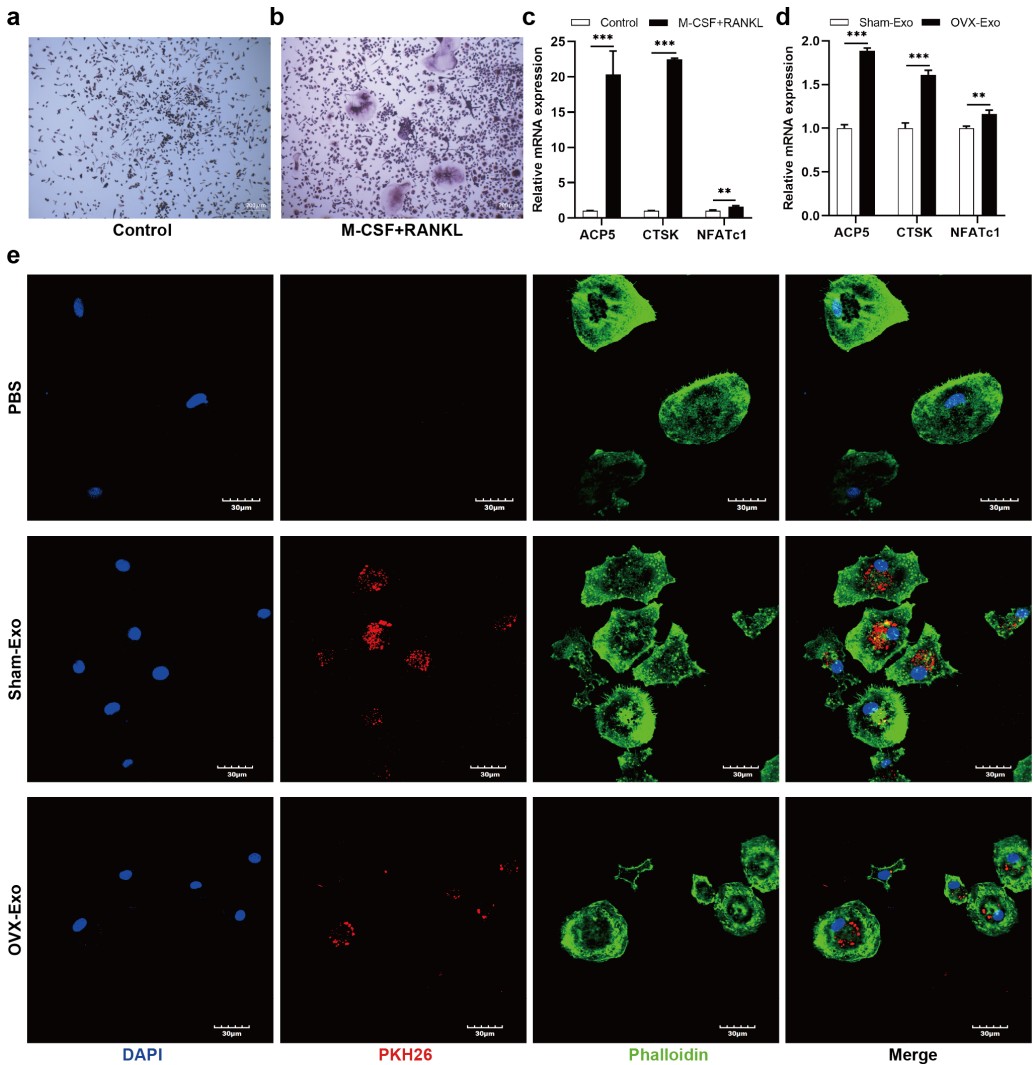

**Figure 3** **OVX-Exo and Sham-Exo are internalized by BMMs, affecting osteoclast differentiation.** (A and B) TRAP staining of the control groups and M-CSF+RANKL induction group (scale bars = 200 μm). (C) The mRNA levels of ACP5, CTSK and NFATc1 in the M-CSF+RANKL induction group compared to the control group determined by qRT-PCR. (D) The mRNA levels of ACP5, CTSK and NFATc1 in osteoclasts cocultured with OVX-Exo compared to Sham-Exo determined by qRT-PCR. (E) Confocal microscopy images show that OVX-Exo and Sham-Exo are internalized by BMMs (scale bars = 30 μm). Exosomes are labeled with PKH26 (red); nuclei are labeled with DAPI (blue); and cytoskeletons are labeled with phalloidin (green). $\beta$-actin is used for normalization of mRNA. Data are presented as the mean ± SD, and all experiments were repeated three times. ** $p < 0.01$, *** $P < 0.001$.

## MiR-27a-3p and miR-196b-5p promoted osteogenic differentiation of BMSCs

To clarify the regulatory roles of these DE-miRNAs on osteogenic differentiation of BMSCs, we selected miR-27a-3p and miR-196b-5p for further study. According to miRBase (http://www.mirbase.org/), the sequence of human miR-27a-3p is identical to that of rat miR-27a-3p, and human miR-196b-5p shared the same sequence of rat miR-196b-5p. The

**Table 1  Differentially expressed miRNAs between OVX-Exo and Sham-Exo.**

| miRNA ID | P-value | Fold-change | Expression level (OVX-Exo vs. Sham-Exo) |
|---|---|---|---|
| Known miRNAs | | | |
| rno-miR-106b-5p | $2.17 \times 10^{-3}$ | [a] | Up |
| rno-miR-31a-5p | $2.68 \times 10^{-3}$ | 3.78 | Up |
| rno-miR-199a-3p | $2.93 \times 10^{-3}$ | 3.46 | Up |
| rno-miR-34c-5p | $3.41 \times 10^{-3}$ | [a] | Up |
| rno-miR-24-3p_R-2 | $6.51 \times 10^{-3}$ | 29.92 | Up |
| mmu-miR-5124a_L+1_1ss6CA | $8.50 \times 10^{-3}$ | 0.37 | Down |
| rno-miR-22-3p | $1.72 \times 10^{-2}$ | 1.77 | Up |
| rno-miR-22-5p_R-1 | $2.13 \times 10^{-2}$ | [a] | Up |
| mmu-miR-744-5p | $2.28 \times 10^{-2}$ | [a] | Up |
| rno-miR-199a-5p | $2.72 \times 10^{-2}$ | 2.09 | Up |
| rno-miR-196b-5p | $3.00 \times 10^{-2}$ | [a] | Up |
| rno-miR-142-5p_L+2R-3 | $3.27 \times 10^{-2}$ | 6.42 | Up |
| rno-miR-146a-5p_R+1 | $4.03 \times 10^{-2}$ | 5.62 | Up |
| rno-miR-27a-3p | $4.69 \times 10^{-2}$ | 4.77 | Up |
| Novel miRNAs | | | |
| PC-3p-47903_116 | $6.43 \times 10^{-3}$ | 0.48 | Down |
| PC-3p-132011_28 | $7.48 \times 10^{-3}$ | 0.37 | Down |
| PC-5p-243764_13 | $1.68 \times 10^{-2}$ | [a] | Up |
| PC-3p-114877_33 | $2.43 \times 10^{-2}$ | 0.16 | Down |
| PC-5p-91335_46 | $4.42 \times 10^{-2}$ | 0.43 | Down |
| PC-5p-9056_653 | $4.70 \times 10^{-2}$ | 0.61 | Down |

**Notes.**

[a]miRNA was not detected in Sham-Exo.

OVX-Exo,  Bone marrow mesenchymal stem cells derived exosomes from ovariectomized rats;  Sham-Exo,  Bone marrow mesenchymal stem cells derived exosomes from sham-operated rats.

expression of miR-27a-3p was decreased in BMSCs derived from OVX rats compared to Sham rats (Fig. 6A). During the osteogenic differentiation of BMSCs, the expression of miR-27a-3p was markedly increased at day 5, 7 and 9 after osteogenesis induction (Fig. 6B). The miR-27a-3p was overexpressed by transfection with miR-27a-3p mimic and knocked down with miR-27a-3p inhibitor. After 12 h of transfection, miR-27a-3p mimic- or miR-27a-3p inhibitor-transfected BMSCs showed significantly higher or lower expression of miR-27a-3p, respectively (Fig. 6C). The transfection reagent was then replaced with osteogenic induction solution. After 48 h of induction, protein and mRNA expression of osteogenic differentiation markers, such as ALP, OCN, OSX and RUNX2, were clearly enhanced in BMSCs transfected with miR-27a-3p mimic compared with the mimic-NC group, whereas the miR-27a-3p inhibitor group showed opposite trend (Figs. 6D–6F). Interestingly, miR-196b-5p was studied in the same way and the results were similar to miR-27a-3p (Figs. 6G–6L).
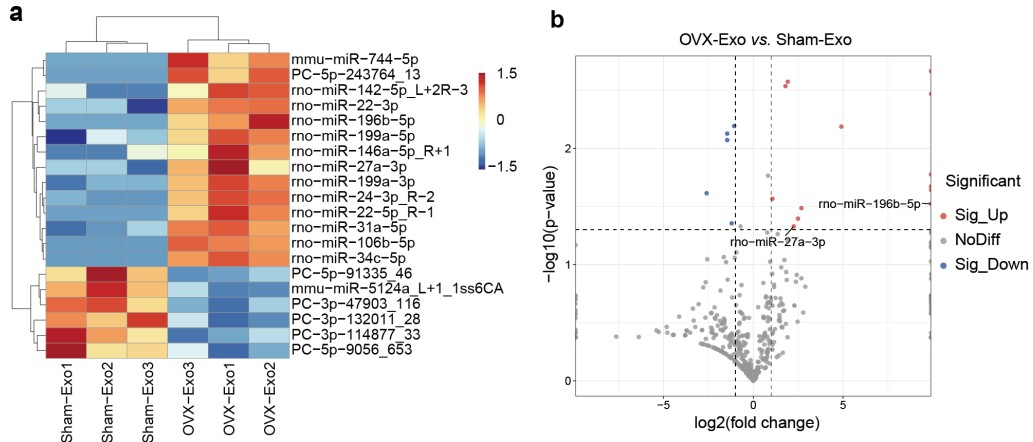

**Figure 4  DE-miRNAs in OVX-Exo and Sham-Exo.** (A) Hierarchical clustering of the DE-miRNAs. Red pixels represent high expression, and blue pixels represent low expression. The original expression values of the miRNAs are normalized using the *Z*-score. The absolute strength of the signal ranges from −1.5 to +1.5, and the corresponding color ranges from blue to red. (B) Volcano map showing the distribution of DE-miRNAs, with rno-miR-27a-3p and rno-miR-196b-5p specially annotated.

## Knockdown of miR-27a-3p and miR-196b-5p promoted osteoclast differentiation

The NGS results were verified by qRT-PCR, indicating that miR-27a-3p and miR-196b-5p were expressed at lower levels in OVX-Exo than that in Sham-Exo (Figs. 7A and 7B). Furthermore, we found that miR-27a-3p and miR-196b-5p expression were decreased during osteoclast differentiation. Briefly, miR-27a-3p and miR-196b-5p expression were decreased at day 4 after osteoclast differentiation induction compared with preinduction (Figs. 7C and 7D). We validated the modulatory effect of miR-27a-3p and miR-196b-5p on osteoclast differentiation by knocking down the expression of miR-27a-3p and miR-196b-5p. The miR-27a-3p inhibitor group had reduced miR-27a-3p expression in osteoclasts (Fig. 7E), while the mRNA expressions of osteoclastic markers ACP5, CTSK and NFATc1 were increased (Fig. 7F). Similarly, the miR-196b-5p inhibitor group had reduced miR-196b-5p expression in osteoclasts (Fig. 7G), while the mRNA expressions of osteoclastic markers were increased (Fig. 7H).

## DISCUSSION

Exosomal miRNA in the bone marrow microenvironment play an important role in the cross-talk between BMSCs and osteoclasts and in the regulation of bone remodeling (*Xu et al., 2018a*; *Xu et al., 2018b*). In our study, we found that OVX-Exo promoted osteoclast differentiation compared to Sham-Exo, while miR-27a-3p and miR-196b-5p were expressed at relatively low levels in OVX-derived BMSCs and OVX-Exo. The expression of osteogenic marker genes increased with increasing levels of miR-27a-3p or miR-196b-5p, whereas knockdown of miR-27a-3p or miR-196b-5p decreased the expression of osteogenic marker genes and promoted the expression of osteoclastic marker genes.
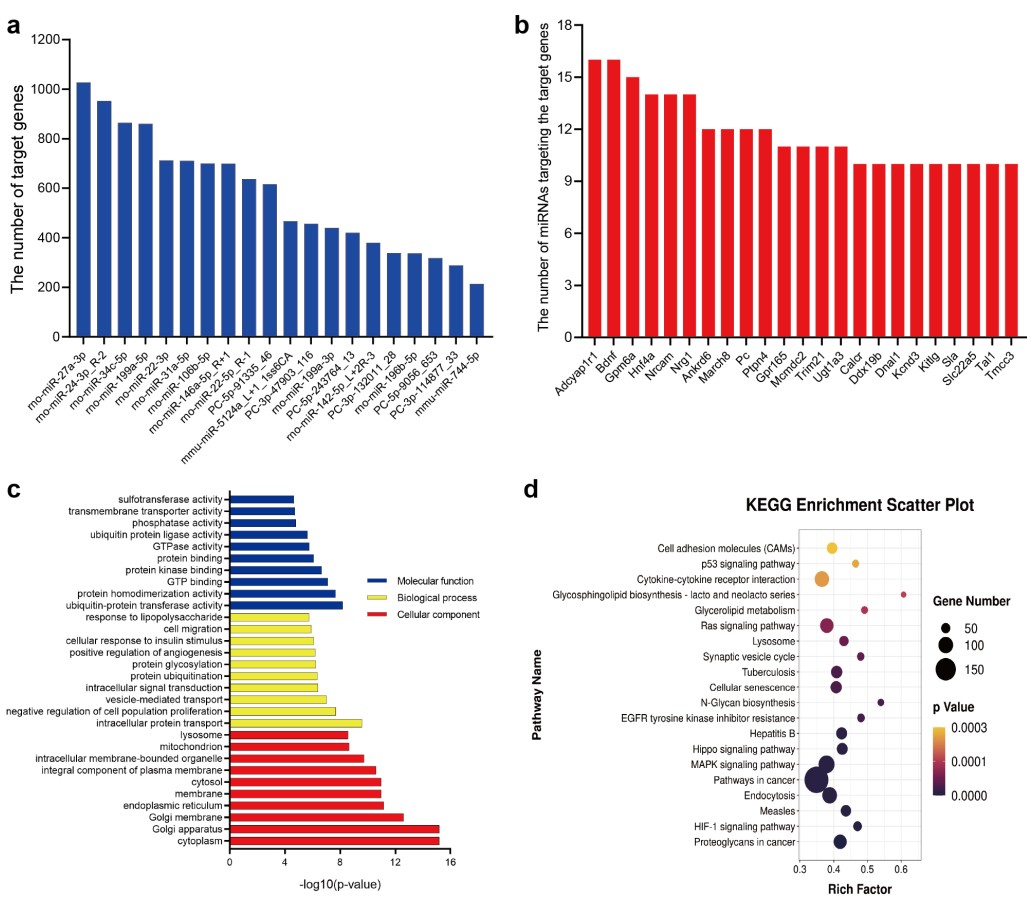

**Figure 5** **Bioinformatics analysis of target genes of DE-miRNAs.** (A) The number of target genes predicted by DE-miRNAs. (B) The number of miRNAs targeting the target genes. (C) GO annotation of target genes. The 10 most enriched GO terms are listed with respect to biological process, cellular component, and molecular function based on *p*-values. (D) The 20 most common KEGG pathways of target mRNAs of DE-miRNAs.

Previously, it has been reported that BMSC-derived supernatant from aged rats was able to promote osteoclast differentiation and bone resorption compared to the young rats (*Xu et al., 2018a*; *Xu et al., 2018b*). However, there are no studies comparing the effects of BMSC-derived exosomes from postmenopausal osteoporotic rats with those of normal rats on osteoclast differentiation. The present study revealed that OVX-Exo promoted osteoclast differentiation compared to Sham-Exo, and this finding has important implications for exploring the role of BMSC-derived exosomes in bone remodeling.

In accordance with the present results, previous studies have demonstrated that miR-27a-3p enhanced osteogenic differentiation of MC3T3-E1 preosteoblasts (*Ren et al., 2021*; *Xu et al., 2018a*; *Xu et al., 2018b*). However, several studies have shown that miR-27a-3p inhibits the osteogenic differentiation of MC3T3-E1 preosteoblasts or BMSCs (*Xu et al., 2020*; *Zeng et al., 2021*; *Furesi et al., 2022*; *Zhang et al., 2017*). The reasons for this discrepancy are unclear and need to be studied in more depth. Comparatively, there are

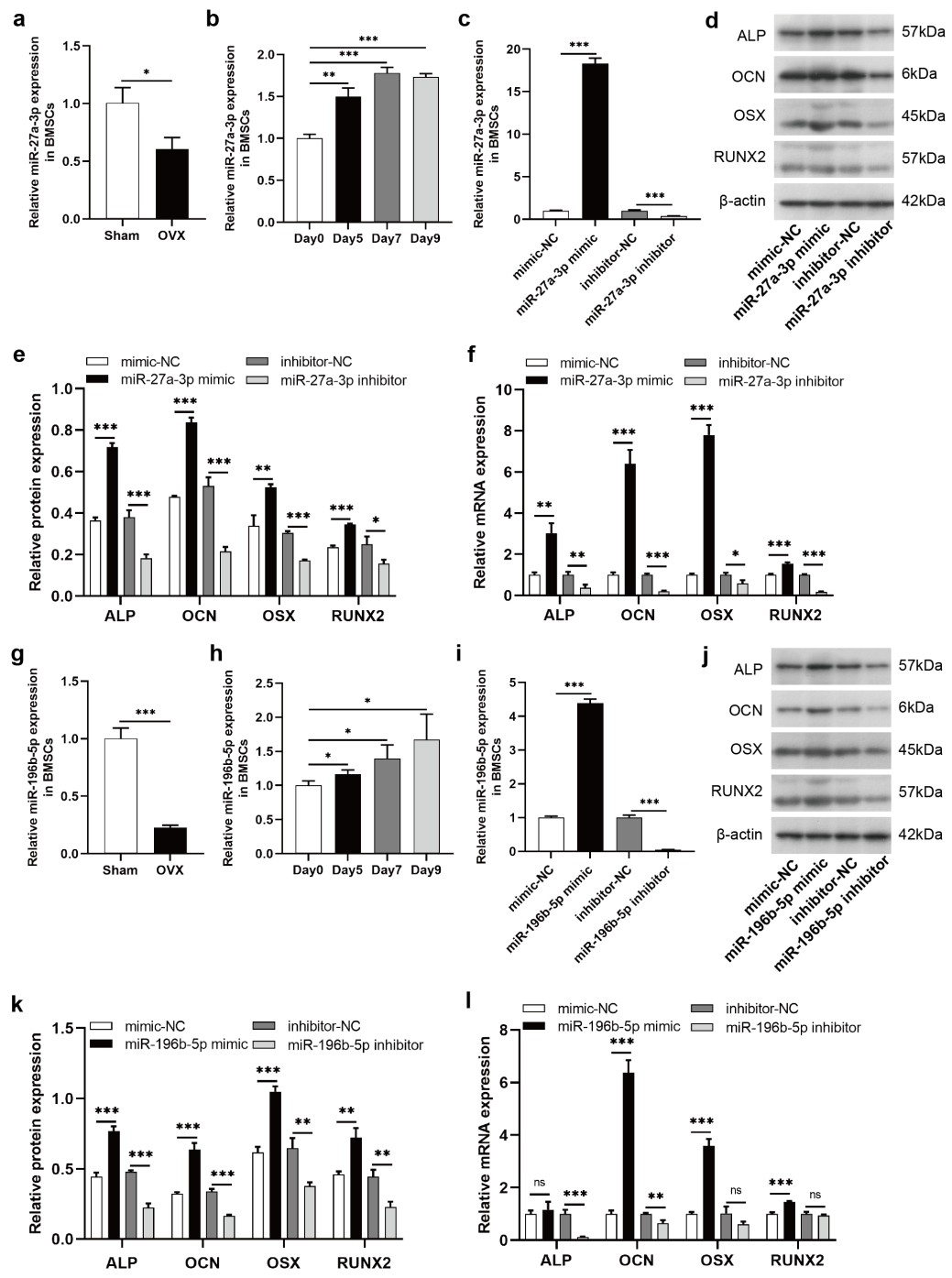

**Figure 6** **MiR-27a-3p and miR-196b-5p promote osteogenic differentiation of BMSCs.** (A) Relative expression of miR-27a-3p in BMSCs from OVX rats compared with Sham rats. (B) Relative expressions of miR-27a-3p in BMSCs on day 5, 7 and 9 after osteogenesis induction compared with preinduction. (C) Relative expressions of miR-27a-3p in BMSCs transfected with miR-27a-3p mimic/inhibitor. (D–F) The protein and mRNA levels of ALP, OCN, OSX and RUNX2 in 

**Figure 6 (…continued)**
BMSCs transfected with miR-27a-3p mimic/inhibitor. (G) Relative expression of miR-196b-5p in BMSCs from OVX rats compared with Sham rats. (H) Relative expressions of miR-196b-5p in BMSCs on day 5, 7 and 9 after osteogenesis induction compared with preinduction. (I) Relative expressions of miR-196b-5p in BMSCs transfected with miR-196b-5p mimic/inhibitor. (J–L) The protein and mRNA levels of ALP, OCN, OSX and RUNX2 in BMSCs transfected with miR-196b-5p mimic/inhibitor. The expression levels of miRNA and mRNA were determined by qRT-PCR, and protein expressions were detected by western blot. U6 is used for the normalization of miRNA and $\beta$-actin is used for normalization of mRNA. $\beta$-actin served as the loading control in the western blot analysis. Data are presented as the mean $\pm$ SD, and all experiments were repeated three times. * $P < 0.05$, ** $p < 0.01$, *** $P < 0.001$, ns: not significant.

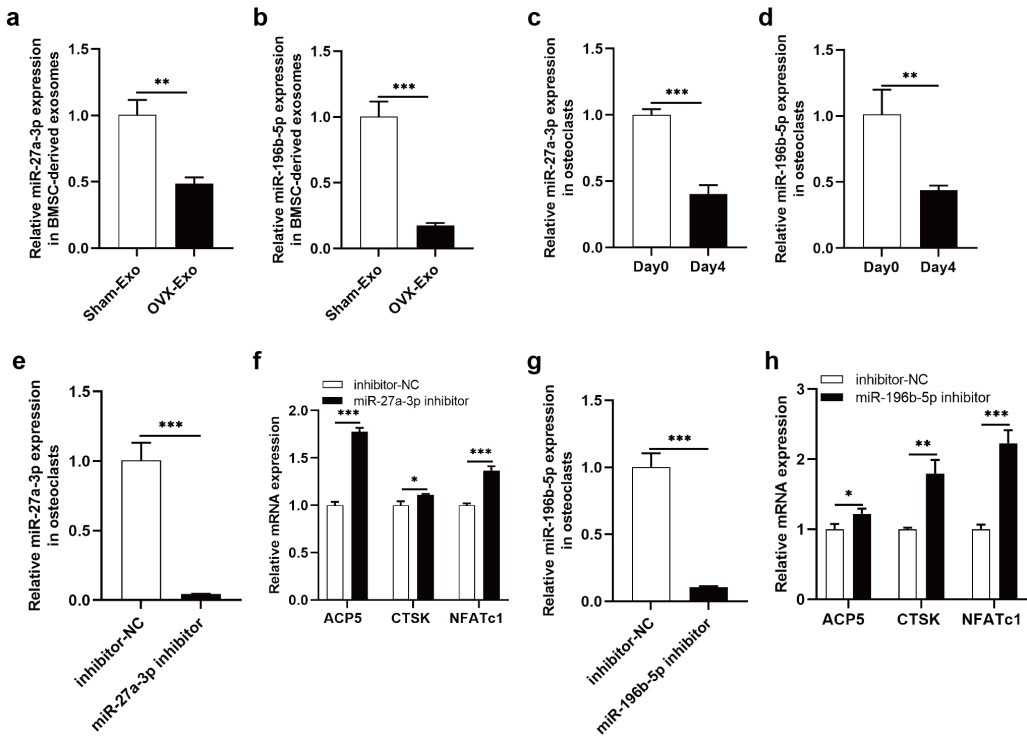

**Figure 7 Knockdown of miR-27a-3p and miR-196b-5p promote osteoclast differentiation.** (A and B) Relative expression of miR-27a-3p and miR-196b-5p in BMSC-derived exosomes from OVX rats compared with Sham rats. (C and D) Relative expression of miR-27a-3p and miR-196b-5p in osteoclasts at day 4 after osteoclast differentiation induction compared with preinduction. (E) Relative expression of miR-27a-3p in osteoclasts transfected with miR-27a-3p inhibitor compared to inhibitor-NC. (F) The mRNA levels of ACP5, CTSK and NFATc1 in osteoclasts transfected with miR-27a-3p inhibitor compared to inhibitor-NC. (G) Relative expression of miR-196b-5p in osteoclasts transfected with miR-27a-3p inhibitor compared to inhibitor-NC. (F) The mRNA levels of ACP5, CTSK and NFATc1 in osteoclasts transfected with miR-196b-5p inhibitor compared to inhibitor-NC. Expression levels were determined by qRT-PCR. Cel-miR-39-3p and U6 are used for exosomal and cellular miRNA normalization, respectively. $\beta$-actin is used for mRNA normalization. Data are presented as the mean $\pm$ SD, and all experiments were repeated three times. * $P < 0.05$, ** $p < 0.01$, *** $P < 0.001$.

few studies on miR-27a-3p regulation of osteoclast differentiation. Ma et al. reported reduced expression of miR-27a-3p during osteoclast differentiation and osteoporosis (*Ma et al., 2016*), and Wang et al. discovered that miR-27a released by MSC derived extracellular

vesicles promotes bone formation and inhibits bone resorption, which is consistent with our findings (*Wang, Zhou & Wang, 2022*).

Some studies have reported that miR-196 families were involved in bone formation and bone resorption, such as miR-196a accelerates osteogenic differentiation in osteoporotic mice and miR-196a-5p in extracellular vesicles secreted by myoblasts inhibits osteoclast-like cell formation (*Zhong et al., 2019*; *Takafuji et al., 2021*). This is consistent with the finding of this study that miR-196b-5p promotes osteoblast differentiation and inhibits osteoclast differentiation. Despite these promising results, questions remain. In osteoporosis, bone formation and bone resorption play equally important roles. Do these miRNAs regulating the osteogenic differentiation of BMSCs modulate their functions through exosomal delivery to osteoclasts? Further research should be undertaken to investigate the specific roles played by these exosomal miRNAs in bone remodeling and their target genes that regulate osteoblast and osteoclast differentiation.

These findings suggest that exosomes act as an important mediator of information exchange between BMSCs and osteoclasts in the bone microenvironment, and exosomal miRNAs can regulate both bone formation and bone resorption. By modifying exosomal miRNAs to enhance osteogenesis while decreasing bone resorption, it may be an effective method for treating osteoporosis.

## CONCLUSIONS

In this study, we found that OVX-Exo promoted osteoclast differentiation compared to Sham-Exo. Overexpression of miR-27a-3p or miR-196b-5p promoted the osteogenic differentiation of BMSCs, and knockdown of miR-27a-3p or miR-196b-5p inhibited the osteogenic differentiation of BMSCs and promoted osteoclast differentiation.

### Funding

This work was supported by Shenzhen Fundamental Research Key Project (No. JCYJ20200109150641992), Shenzhen Science and Technology Innovation Committee (No. JCYJ20170818164059405), the Key-Area Research and Development Program of Guangdong Province, China (No. 2020B010165004) and Category A of Shenzhen Hong Kong Jointly Funded Project (No. SGDX20201103095600002). The funders had no role in study design, data collection and analysis, decision to publish, or preparation of the manuscript.

### Grant Disclosures

The following grant information was disclosed by the authors:
Shenzhen Fundamental Research Key Project: JCYJ20200109150641992.
Shenzhen Science and Technology Innovation Committee: JCYJ20170818164059405.
Key-Area Research and Development Program of Guangdong Province, China: 2020B010165004.
Category A of Shenzhen Hong Kong Jointly: SGDX20201103095600002.

## Competing Interests

The authors declare there are no competing interests.

## Author Contributions

- Guohua Lai conceived and designed the experiments, performed the experiments, analyzed the data, prepared figures and/or tables, authored or reviewed drafts of the article, and approved the final draft.
- Renli Zhao performed the experiments, prepared figures and/or tables, and approved the final draft.
- Weida Zhuang analyzed the data, authored or reviewed drafts of the article, and approved the final draft.
- Zuoxu Hou conceived and designed the experiments, analyzed the data, authored or reviewed drafts of the article, and approved the final draft.
- Zefeng Yang performed the experiments, prepared figures and/or tables, and approved the final draft.
- Peipei He performed the experiments, prepared figures and/or tables, and approved the final draft.
- Jiachang Wu conceived and designed the experiments, authored or reviewed drafts of the article, and approved the final draft.
- Hongxun Sang conceived and designed the experiments, authored or reviewed drafts of the article, and approved the final draft.

## Animal Ethics

The following information was supplied relating to ethical approvals (*i.e.*, approving body and any reference numbers):

All experimental animal procedures were approved by the Animal Ethics Committee of Shenzhen Hospital of Southern Medical University, China (approval No. 2021-0057).

## DNA Deposition

The following information was supplied regarding the deposition of DNA sequences:

The sequences are available at NCBI GEO: GSE197085.

Enter token "atezqkqabpwlbmv" into the box to gain access.

## Data Availability

The raw data is available at Zenodo: Lai, Guohua. (2022). Raw data [Data set]. Zenodo. https://doi.org/10.5281/zenodo.6592255.

## Supplemental Information

Supplemental information for this article can be found online at http://dx.doi.org/10.7717/peerj.13744#supplemental-information.

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
