# Peer review of "BMSC-derived exosomal miR-27a-3p and miR-196b-5p regulate bone remodeling in ovariectomized rats"

_PeerJ, doi:10.7717/peerj.13744_

## Round 0.1 · original submission · Major Revisions

Make sure to follow all reviewer´s suggestions. Specially, restructure the whole discussion section as suggested, include a negative marker of exosomes, correct errors in figure order and include the sample size in figure legend and in the manuscript sections (whenever relevant).

Reviewer 1 ·

Basic reporting

In the paper entitled "BMSC-derived exosomal miR-27a-3p and miR-196b-5p regulate bone remodeling in ovariectomized rats", the authors studied BMSC-derived exosomes from ovariectomized rats could enhance osteoclast differentiation, and miR-27a-3p and miR-196b-5p were identified as key exosomal MiRs.The manuscript is well written.

Experimental design

The in vitro data are robust and consistent with previous reports. However, there are some issues that need to be further developed.
1.Line 258. The authors should set up exosome negative marker proteins, not only postive markers.
2.Line 269. The authors should add a blank contrast group such as PBS group to confirm the effects of exosomes
3.Line 304. The authors should provide Western Blot data to show miR-27a-3p could affect the expression of osteogenic genes. The protein datas could make the results more credible.

Validity of the findings

The data are validated, conclusion is convincing.

Additional comments

In Figure 1, Figure 6 and Figure 7, there were problems with the order of the figure legends,the authors need to rearrange the order of the figures.
In Figure 4a, where are miR-27a-3p and miR-196b-5p and Maybe an arrow could show it. It is good for the readers to know where the candidate is in the volcano plot.
In the Results of the Abstract, “osteoclast s” should be “osteoclasts”

Reviewer 2 ·

Basic reporting

This is a manuscript of great interest in which the authors identify and associate two miRNAs, naturally transported within bone marrow mesenchymal stem cell (BMSC)-derived extracellular vesicles, with osteoclast differentiation.
The study by Guohua Lai and colleagues shows promising results of the potential use of miRNAs transported within extracellular vesicles providing further information for RNA-based therapies.
The paper is well organized and I find the experiments well designed. Authors corroborate their results from different points of view using miRNA mimics and miRNA inhibitors to measure the expression of some genes that participate in both, osteogenic and osteoclast differentiation.
However, the way in which the topic is introduced, the way in which the results are described and, above all, the way in which the discussion is developed should be improved. Authors do not reflect the importance of their results obtained.

Experimental design

- Introduction:
The introduction is too simplified. I believe that the authors could reduce the information provided between lines 77 and 88 and introduce some of the parameters employed to detect osteoporosis (ROI, BMD, tb. N etc.) as well as the genes that are over-expressed or under-expressed during the disease to better understand the experiments they perform. In addition, the authors described how they performed the experiment but not clearly explain the ultimate goal they intend to pursue.
- Materials and methods:
Describe the rotor and centrifuge type/characteristics employed for ultracentrifugation.
- Results:
Mention the results according to the order in which they appear in the figures.
For example, on line 251, put the surface markers of MSCs in the same order as they appear in the figure. The same must be done when the authors describe OVX and sham during all results (Figure 1 and 3). In addition, the legend in Figure 1 should be in alphabetical order (subfigure "d" cannot be explained before "b or c").
- Discussion:
As presented, the discussion cannot be considered as such. The authors do not discuss their results nor compare them with the existing literature, they just write a list of articles, mentioned one after another, talking about miRNAs involved in the regulation of genes associated with osteogenic and osteoclast differentiation. Many papers are cited and not are carefully discussed. Please, rewrite the discussion trying to avoid the shopping list of papers, instead carefully discuss your results and compare them with the current literature.

Validity of the findings

XX

Additional comments

NGS analysis revealed that the miRNA rno-miR-196b-5p was only expressed in OVX-Exo. However, when the authors verified the NGS results by qRT-PCR, they detected that rno-miR-196b-5p was over-expressed up to 5-fold more in sham-Exo samples compared to OVX-Exo (line 312 and Figure 7e). How can the authors explain this result? It is possible that I am not understanding it properly so please, explain it better taking into account the results obtained from NGS and qRT-PCR or modify it according to the results.
Suggestions: According to the International Society of Extracellular Vesicles (ISEV) at least one protein absence in exosomes (e.g. calnexin or cytochrome C) should be examined simultaneously with proteins characterized in exosomes, such as C9, CD63 and CD81, to rule out the presence of cellular contamination in the preparations. Please, take this recommendation into account for the following experiments. [http://dx.doi.org/10.3402/jev.v3.26913]
It is important to mention in the text that rat miRNAs selected for the study (rno-miR-27a-3p and rno-miR-196b-5p) share the same sequence to human miRNAs.

Reviewer 3 ·

Basic reporting

In this manuscript,  the authors found that miR-27a-3p and miR-196b-5p promoted BMSCs osteogenic differentiation and knockdown of BMSC-derived exosomal miR-27a-3p and miR-196b-5p promoted osteoclast differentiation.

Experimental design

The specific experimental operation was not detailed, and the postoperative treatment was not described.The article failed to state the number of sample sizes used for the experimental results.

Validity of the findings

The magnification of transmission electron microscopy and trap standing image is inconsistent.
The original image is not used in Western blot, and the image modification trace is obvious.
The background color of micro CT three dimensional view is inconsistent, and the modification trace is obvious.

Additional comments

There were some problems with the order of the figure legends,the authors need to rearrange the order of the figures.

---

## Round 0.2 · Minor Revisions

Authors have modified the manuscript according to reviewer's suggestions. Although some minor points need to be revised (see reviewer comments).

Reviewer 2 ·

Basic reporting

See bellow

Experimental design

See bellow

Validity of the findings

See bellow

Additional comments

The authors have included and corrected the comments made in the previous review.
Some details to change:
- Line 285: “Analysis of morphological data showed that BMD, Tb. N, BV/TV and BS/TV of the fourth lumbar vertebrae were decreased in OVX rats compared to Sham rats”. However, no statistical differences have been observed (Figure 1a) for BV/TV and BS/TV. Authors should clarify that a trend is observed but cannot conclude that these two parameters decreased.
- Line 322: “NGS revealed that a total of 708 miRNAs were identified in OVX-Exo and Sham Exo (Supplementary Table 2). Among these, 20 miRNAs (14 known and 6 novel) 324 were significantly and differentially expressed…”. However, only 17 known miRNAs are shown in Table 1. Moreover, not all of them are included in the Hierarchical clustering.
- Authors must explain the concentration of miRNA mimic and miRNA inhibitor employed for cell transfection.
- In figure 7 the legend on the X-axis of some graphs is missing.

---

## Round 0.3 · accepted · Accept

Although authors have used very high concentrations of miRNAs mimics and inhibitors for gain and loss of function activities (which is a concern in this field), they have adequately responded to the suggestions raised by reviewers.